# The Role of Emerging Immune-Inflammatory Indexes in the Preoperative Differentiation of Complicated and Uncomplicated Acute Appendicitis: A Single-Center Retrospective Analysis

**DOI:** 10.3390/diagnostics16010021

**Published:** 2025-12-20

**Authors:** Botond-István Kiss, Daniela-Tatiana Sala, Renáta Moriczi, Szabolcs-Attila Gábor, Árpád Török, Tivadar Bara, Mircea-Gabriel Mureșan, Valentin Daniealopol, Szilárd-Leó Kiss, Radu-Mircea Neagoe

**Affiliations:** 1Doctoral School of Medicine and Pharmacy, “George Emil Palade” University of Medicine, Pharmacy, Science and Technology of Târgu Mureș, 540139 Târgu Mureș, Romania; drkissbotondi@gmail.com (B.-I.K.); moriczi.renata@yahoo.com (R.M.); valentin.daniealopol@yahoo.com (V.D.); 22nd Department of Surgery, “George Emil Palade” University of Medicine, Pharmacy, Science and Technology of Târgu Mureș, 540139 Târgu Mureș, Romania; 3Department of Anatomy and Embryology, “George Emil Palade” University of Medicine, Pharmacy, Science and Technology of Târgu Mureș, 540139 Târgu Mureș, Romania; 42nd Clinic of Surgery, Mureș County Emergency Clinical Hospital, 540136 Târgu Mureș, Romania; gabor.szabi1@yahoo.com (S.-A.G.); torokaea@gmail.com (Á.T.); btibi_ms@yahoo.com (T.B.); mircea.muresan@umfst.ro (M.-G.M.); neagoerm@gmail.com (R.-M.N.); 51st Clinic of Gynecology and Obstetrics, Mureș County Emergency Clinical Hospital, 540136 Târgu Mureș, Romania; k.szilardleo@iclould.com

**Keywords:** acute appendicitis, complicated acute appendicitis, preoperative risk stratification, immune-inflammation biomarkers, ROC curve, length of stay, costs

## Abstract

**Background/Objectives**: Acute appendicitis (AA) is among the most common surgical emergencies. Differentiating between complicated (CAA) and uncomplicated (UAA) forms is essential for selecting the appropriate management—operative or non-operative—and for optimizing patient prioritization and outcomes. This study aimed to evaluate the diagnostic performance of emerging inflammatory indices in distinguishing these forms of AA. **Methods**: A total of 514 adult patients with surgically confirmed AA were retrospectively analyzed. Six immune-inflammatory indices—neutrophil-to-lymphocyte ratio (NLR), platelet-to-lymphocyte ratio (PLR), monocyte-to-lymphocyte ratio (MLR), systemic immune-inflammation index (SII), systemic inflammation response index (SIRI), and pan-immune-inflammation value (PIV)—were calculated and compared with intraoperative and histopathological findings. Postoperative outcomes, including length of hospital stay (LOS) and hospitalization costs, were also evaluated. **Results**: All six indices were significantly higher in intraoperatively identified complicated cases (*p* < 0.0001). In histopathological analysis, five indices (NLR, MLR, SII, SIRI, and PIV) remained significantly elevated in patients with wall necrosis or perforation (*p* = 0.000–0.019), while PLR did not reach statistical significance. The indices showed fair diagnostic accuracy (AUC = 0.664–0.719, *p* < 0.0001). NLR and MLR were independent risk factors for CAA (*p* = 0.006 and *p* = 0.016), and MLR was also independently associated with complicated histopathological findings (*p* = 0.036). PIV independently predicted both increased LOS and higher hospitalization costs (*p* = 0.001 for each). **Conclusions:** These easily calculable inflammatory markers can serve as useful adjuncts for preoperative stratification of AA, supporting timely decision-making and contributing to more cost-effective emergency surgical care.

## 1. Introduction

Acute appendicitis (AA) is among the most frequent pathologies encountered in the emergency department requiring abdominal surgery. Prompt and adequate treatment is essential to keep morbidity and mortality low [1]. AA is commonly classified into uncomplicated and complicated forms, based on the extent of inflammation and the presence of local or generalized septic complications [2]. Uncomplicated appendicitis (UAA) is limited to phlegmonous inflammation without necrosis or perforation, whereas complicated appendicitis (CAA) involves transmural necrosis, perforation, or diffuse peritonitis [3]. This distinction has major therapeutic implications: antibiotic-based non-operative management has emerged as a safe option for selected uncomplicated cases, while surgical intervention remains the treatment of choice for complicated disease [4,5]. Consequently, accurate preoperative stratification between these entities is essential to guide management decisions and optimize patient outcomes [3].

Recently, new inflammatory indices have emerged to assist clinicians in differentiating between complicated and uncomplicated cases. The neutrophil-to-lymphocyte ratio (NLR; neutrophil/lymphocyte) is a specific and easily obtainable inflammatory marker derived from routine peripheral blood counts. Elevated NLR values are frequently associated with severe inflammation and reflect the balance between two major leukocyte lineages: neutrophils (innate immunity) and lymphocytes (adaptive immunity) [1,6]. The platelet-to-lymphocyte ratio (PLR) is another composite biomarker that integrates inflammatory and prothrombotic pathways of the systemic response [7]. The monocyte-to-lymphocyte ratio (MLR) has also gained attention as a cost-effective marker of immune dysregulation, reflecting both activation of innate immune monocytes and suppression of adaptive lymphocytes. Increased MLR levels have been correlated with the severity of acute inflammatory and infectious conditions, including appendicitis [8,9]. Beyond these simpler ratios, several emerging multiparameter indices have been proposed to better capture the complexity of systemic immune activation. The Systemic Immune-Inflammation Index (SII; platelet × neutrophil/lymphocyte) and the Systemic Inflammation Response Index (SIRI; neutrophil × monocyte/lymphocyte) have also shown promising early results in differentiating complicated from UAA [10,11]. The Pan-Immune-Inflammation Value (PIV; neutrophil × platelet × monocyte/lymphocyte) extends this concept by incorporating all four major circulating immune cell lineages. Although PIV has been validated in oncologic and systemic inflammatory contexts, its diagnostic role in AA remains insufficiently explored and requires further clinical investigation [12,13]. Several laboratory markers and clinical scores have already been proposed to distinguish UAA from CAA. Hyponatremia, procalcitonin, the delta neutrophil index (DNI) and hyperbilirubinemia have all been associated with complicated disease in adult and pediatric patients, while clinical tools such as the Appendicitis Inflammatory Response (AIR) score combine symptoms, signs and basic labs to stratify risk. Recent work has also highlighted salivary biomarkers as a promising non-invasive adjunct in pediatric appendicitis, although evidence remains limited. Within this landscape, it is still unclear whether simple immune–inflammation indices derived from routine blood counts can provide additional, easily accessible information for preoperative risk stratification [14,15,16,17,18].

The objective of this study was to investigate the diagnostic accuracy and prognostic relevance of emerging immune-inflammatory indices (NLR, PLR, MLR, SIRI, SII, and PIV) in AA. Despite growing evidence supporting their clinical utility, their exact role in preoperative stratification and outcome prediction has not yet been universally established.

## 2. Materials and Methods

### 2.1. Study Design and Setting

This retrospective, single-center study was conducted at the 2nd Department of Surgery, Emergency County Clinical Hospital of Targu Mures, Romania, a university-affiliated teaching hospital. Data were retrieved from the institutional electronic database (H3 Healthcare Concept System). The study design and reporting followed the STROBE (Strengthening the Reporting of Observational Studies in Epidemiology) guidelines.

### 2.2. Study Population

Inclusion criteria were: age ≥ 18 years, emergency appendectomy for clinically suspected AA, intraoperative findings compatible with AA (hyperemia, edema, fibrinous or purulent exudate, gangrene or perforation), and histopathological confirmation of AA.

Exclusion criteria were: negative appendectomy (normal intraoperative and histopathological findings), incidental appendectomy, appendiceal neoplasms and cases with discordant findings (histopathological confirmation of AA in the absence of any macroscopic inflammatory changes intraoperatively).

A total of 514 patients were included during a 7-year period (2018–2024). For each eligible patient, demographic, clinical, preoperative (imaging and laboratory), intraoperative, postoperative and histopathological data were collected from the electronic medical record, including age, sex, residential status (urban/rural), intraoperative diagnosis (uncomplicated vs. complicated AA), preoperative routine laboratory parameters required to calculate the immune–inflammation indices (complete blood count with differentials and C-reactive protein), appendiceal diameter on preoperative CT (when performed), histopathological findings including the presence or absence of wall necrosis, ulceration or perforation (NUP status), operative time, postoperative length of hospital stay and total hospital costs. Preoperative laboratory values were taken from the first blood sample obtained at emergency department presentation, before hospital admission and prior to any analgesic or antibiotic therapy.

The study database was created by retrospectively identifying all consecutive adult patients undergoing emergency appendectomy for suspected AA who fulfilled the predefined inclusion and exclusion criteria. Patients with negative or incidental appendectomy, appendiceal neoplasms, or discordant histopathological findings were excluded at the time of database construction. As these ineligible cases were not systematically logged, their exact number could not be determined retrospectively; this process is summarized in the patient flow diagram (Figure 1).

### 2.3. Definition of Complicated Appendicitis

Complicated appendicitis was defined intraoperatively as the presence of perforation, gangrenous (full-thickness) necrosis, intra-abdominal abscess, or localized/generalized peritonitis.

### 2.4. Surgical Technique and Subgroup Classification

Patients were divided according to surgical approach into laparoscopic and converted/open groups. Conversion or primary open appendectomy was performed via a midline laparotomy, including peritoneal exploration, appendectomy, lavage, and drainage when indicated.

Following histopathological examination, patients were further stratified into two subgroups:NUP group (necrosis–ulceration–perforation): cases with full-thickness necrosis, partial necrosis, mucosal ulceration, or perforation;Control group: cases without wall necrosis or perforation.

### 2.5. Study Outcomes

The primary outcome was the presence of complicated acute appendicitis, defined by intraoperative findings (gangrene, perforation, abscess or diffuse peritonitis) versus uncomplicated disease. The main objective was to evaluate the diagnostic performance of immune–inflammatory indices (NLR, PLR, SII, SIRI, MLR, PIV) for distinguishing complicated from uncomplicated appendicitis. Secondary outcomes were: (1) the association of these indices with histopathological evidence of wall necrosis, ulceration or perforation (NUP status), and (2) their relationship with postoperative length of hospital stay and total hospital costs.

### 2.6. Data Management and Statistical Analysis

All data were entered into Microsoft Excel and analyzed using IBM SPSS Statistics version 25.0 (IBM Corp., Armonk, NY, USA).

Categorical variables were compared using the chi-square test. Outliers were identified using Grubbs’ test. Levene’s test was used to assess the equality of variances, and the Shapiro–Wilk test evaluated the normality of continuous variables.

For normally distributed data, comparisons between groups were performed using Student’s *t*-test; for non-normally distributed data, the Mann–Whitney U test was applied. A *p*-value < 0.05 was considered statistically significant. Missing data were handled using a complete-case approach. For each analysis, patients with missing values for the specific variable of interest were excluded only from that particular analysis, while remaining eligible for all other analyses in which their data were available. No imputation of missing values was performed.

After ROC curve and AUC analysis, optimal cut-off values were obtained using Youden’s index without internal validation and should therefore be considered exploratory.

In the regression tables, B denotes unstandardized coefficients with their 95% confidence intervals and standard errors (SE); β represents standardized coefficients. The t and *p* values test the null hypothesis that B = 0. Model fit was assessed by R^2^, adjusted R^2^, and the F statistic. The change in explained variance after addition of inflammatory markers was reported as ΔR^2^ with its corresponding F-change test. Multicollinearity was evaluated using the variance inflation factor (VIF). For bivariate correlations, Spearman’s test was used for non-normally distributed data.

A stepwise binary logistic regression (Forward: LR method) was performed in SPSS to determine which inflammatory markers were independent risk factors for complicated appendicitis. Variables with a *p*-value < 0.05 in univariate analysis were entered into the model. Results were reported as odds ratios (OR) with corresponding 95% confidence intervals (CI). Due to the right-skewed distribution of the data, both length of stay (LOS) and costs were logarithmically transformed prior to multiple linear regression analyses.

## 3. Results

### 3.1. General and Intraoperative Characteristics of the Study Population

A total of 514 patients underwent surgery for AA, including 224 females (43.6%) and 290 males (56.4%). Catarrhal appendicitis was identified in 40 cases (7.8%), phlegmonous in 214 cases (41.6%), and gangrenous in 237 cases (46.1%). Generalized peritonitis was present in 93 patients (18.3%), periappendicular or pericecal abscess in 77 (15.0%), localized peritonitis in 101 (19.6%), and perforated appendicitis in 149 (29.0%). Overall, 268 patients (52.1%) had CAA, while 246 (47.9%) had uncomplicated disease. Laparoscopic appendectomy was performed in 422 cases (82.1%); conversion to open surgery was required in 50 cases (9.7%), and a primary open approach was used in 43 cases (8.4%).

### 3.2. Preoperative Laboratory and Clinical Differences Between Study Groups

Preoperative emerging inflammatory markers showed significant differences between the complicated and uncomplicated groups (Table 1). Patients with CAA were significantly older than those with uncomplicated disease (46.28 ± 18.06 vs. 35.03 ± 14.66 years; *p* = 0.0001), had longer operative times (79.78 ± 26.25 vs. 60.75 ± 20.999 min; *p* = 0.0001), and had longer hospital stay than those with uncomplicated disease (median 4 [3.0–6.0] vs. 3 [2.0–4.0] days, Z = −9.688, *p* < 0.0001). CT-measured appendiceal diameter was greater in complicated cases (median 13 mm [IQR 10–15] vs. 10 mm [8.15–12]; *p* < 0.0001). All six emerging immune-inflammatory indices demonstrated significantly higher median values in patients with CAA than in those with uncomplicated disease (*p* < 0.0001 for all comparisons). Among them, NLR, PLR, and SII showed the most pronounced intergroup differences, with median values nearly doubled in the complicated subgroup. Leukocyte, neutrophil, and monocyte counts were also higher in the complicated group (all *p* ≤ 0.001), lymphocyte counts were lower (median 1.31 vs. 1.86 × 10^9^/L; *p* = 0.0001), while platelet counts showed no significant difference (*p* = 0.249).

### 3.3. Association of Immune-Inflammatory Indices with Histopathological Severity of Appendicitis

Patients with histopathological findings of necrosis, ulceration, or perforation (NUP-positive group) demonstrated significantly higher values of several immune–inflammatory markers compared with those without these changes (Table 2). Median NLR was markedly increased in the NUP-positive group (8.58 [IQR 5.65–11.82]) versus NUP-negative cases (6.16 [IQR 3.95–11.18]; *p* < 0.001, Z = −3.76). Similarly, SII (2030.9 [IQR 1394.2–2952.7] vs. 1600.9 [IQR 1015.9–2785.4]; *p* = 0.010, Z = −2.58), MLR (0.70 [IQR 0.50–0.79] vs. 0.53 [IQR 0.35–0.79]; *p* = 0.001, Z = −3.22), SIRI (7.35 [IQR 4.47–11.65] vs. 5.67 [IQR 2.56–9.93]; *p* = 0.002, Z = −3.08), and PIV (1767.9 [IQR 1110.9–2899.2] vs. 1484.3 [IQR 626.7–2457.9]; *p* = 0.019, Z = −2.34) were all significantly elevated among specimens showing histological wall destruction. In contrast, PLR values did not differ significantly between groups (*p* = 0.235). These results indicate that NLR, MLR, SII, SIRI, and PIV are associated with microscopic tissue necrosis and inflammatory injury, suggesting their potential role as histopathological severity indicators in AA.

### 3.4. Diagnostic Performance of Emerging Immune-Inflammatory Indices: ROC Curve Analysis

ROC curve analysis (Figure 1, Table 3) revealed that all indices had acceptable discriminatory ability for identifying CAA, with AUC values ranging from 0.664 (PLR) to 0.719 (NLR). NLR achieved the highest overall diagnostic performance (AUC = 0.719; sensitivity = 0.866; specificity = 0.481; accuracy = 0.678), followed by MLR (AUC = 0.712) and SIRI (AUC = 0.707). The optimal cut-off points determined by the Youden index were 5.05 for NLR, 138.52 for PLR, 1300.39 for SII, 0.537 for MLR, 4.60 for SIRI, and 1279.43 for PIV. Positive predictive values (PPV) ranged from 0.625 to 0.702, while negative predictive values (NPV) remained consistently above 0.66 across all indices, indicating reliable exclusion capacity for non-complicated cases.

As illustrated in Figure 2, all six ROC curves lie clearly above the reference line, supporting their overall diagnostic value. The curves for NLR, MLR, and SIRI are positioned closer to the upper-left corner, indicating better discriminative performance compared with the other indices. These visual findings are consistent with the regression analysis, in which NLR and MLR remained independent predictors of CAA, further emphasizing their robustness and clinical relevance in the emergency diagnostic setting.

### 3.5. Multivariate Logistic Regression Analysis of Predictive Factors for Complicated Appendicitis

Stepwise binary logistic regression identified the neutrophil-to-lymphocyte ratio (NLR) and the monocyte-to-lymphocyte ratio (MLR) as independent predictors of complicated appendicitis (Table 4). In the first step, NLR alone entered the model (B = 0.125, *p* < 0.001), corresponding to an odds ratio (OR) of 1.134 (95% CI, 1.087–1.183). After inclusion of MLR, both parameters remained significant: NLR (B = 0.078, *p* = 0.006; OR = 1.081; 95% CI, 1.023–1.142) and MLR (B = 1.029, *p* = 0.016; OR = 2.798; 95% CI, 1.212–6.459). These findings indicate that increasing values of NLR and MLR are independently associated with higher odds of CAA, while other indices (PLR, SII, SIRI, PIV) did not contribute significantly to the final predictive model.

In the logistic regression model evaluating histopathological severity (wall necrosis, ulceration, or perforation), MLR emerged as the sole independent predictor of worse microscopic findings (Table 4). Higher MLR values were significantly associated with advanced tissue damage (B = 0.542, *p* = 0.036), corresponding to an OR of 1.719 (95% CI, 1.035–2.856). This indicates that each unit increase in MLR confers a 1.7-fold higher likelihood of complicated histopathological findings. No other inflammatory indices (NLR, PLR, SII, SIRI, or PIV) reached statistical significance in this model.

### 3.6. Hierarchical Multiple Linear Regression Analysis of Inflammatory Indices and Clinical Outcomes

Two hierarchical multiple linear regression models were developed to assess the independent associations between inflammatory indices and clinical outcomes (Table 5). To mitigate multicollinearity among the six inflammatory indices, we restricted the hierarchical models to PIV and PLR. These two markers demonstrated the lowest intercorrelation (i.e., acceptable collinearity diagnostics relative to the others) and were therefore retained together in the final analyses.

In the first model, the log-transformed hospital length of stay (LN_LOS) served as the dependent variable. Age, sex, and residential status (Block 1) explained 24.5% of the variance (adjusted R^2^ = 0.220; *p* < 0.001). Adding PLR and PIV (Block 2) significantly improved model fit (ΔR^2^ = 0.137; F-change [2, 89] *p* < 0.001), increasing the total explained variance to 38.2% (adjusted R^2^ = 0.347). Within the final model, PIV remained an independent predictor of longer hospitalization (B = 0.0008 [95% CI, 0.000–0.000]; β = 0.326; *p* = 0.001), whereas PLR was not significant (*p* = 0.303).

The second model used log-transformed hospital cost (LN_COSTS) as the outcome. Demographic variables accounted for 17.0% of the variance (adjusted R^2^ = 0.143; *p* = 0.001), and the inclusion of PLR and PIV raised the explained variance to 32.9% (adjusted R^2^ = 0.292; ΔR^2^ = 0.160; F-change [2, 89] = 10.589; *p* < 0.001). Again, only PIV was a statistically significant independent predictor of increased cost (β = 0.315; *p* = 0.001), whereas PLR was not (*p* = 0.107).

Multicollinearity diagnostics were satisfactory (VIF = 1.19–1.21 for all variables). Taken together, these findings indicate that higher PIVs are consistently associated with both prolonged hospitalization and increased economic burden, even after adjustment for demographic factors.

## 4. Discussion

AA remains among the most frequent indications for emergency surgery worldwide, with appendectomy—open or laparoscopic—historically regarded as the standard of care [2]. AA is now classified in clinical guidelines into uncomplicated (UAA) and complicated (CAA) forms according to the extent of inflammatory involvement [2]. CAA is associated with substantially higher morbidity and mortality—driven by perforation, abscess formation, or generalized peritonitis; therefore, prompt surgical management remains essential [2,6]. According to current international guidelines, non-operative (conservative) management is recommended exclusively for selected patients with UAA, whereas surgical intervention remains the standard of care for complicated forms [2,19]. Despite numerous clinical and imaging approaches, a universally accepted protocol for the preoperative diagnosis of CAA has not yet been established [20].

Contemporary guidelines endorse non-operative management (NOM) with antibiotics as a safe alternative in selected cases of uncomplicated AA, provided patients are counseled regarding failure/recurrence risk [2,5]. Randomized and pragmatic trials (APPAC, CODA) demonstrate that antibiotics can achieve non-inferior short-term health status versus appendectomy, although a substantial proportion require delayed surgery over time (≈30% by 90 days in CODA; ≈39% recurrence by 5 years in APPAC) [4,5,21]. This evolving therapeutic latitude makes accurate preoperative differentiation of CAA versus UAA pivotal—both to avoid undertreating perforated/necrotic disease and to permit antibiotic-first strategies where appropriate [2,5]. In elderly or frail patients with significant cardiopulmonary comorbidity, NOM may mitigate perioperative risk, but selection must be imaging-guided and individualized given different failure and mortality profiles in this population [19,22,23]. Beyond clinical safety, multiple economic analyses suggest antibiotic-first pathways can be cost-effective for UAA—an important consideration for smaller or resource-limited hospitals—reinforcing the value of robust, reproducible diagnostic algorithms that separate CAA from UAA at presentation [24,25].

Among inflammation-based markers, the neutrophil-to-lymphocyte ratio (NLR) and platelet-to-lymphocyte ratio (PLR) remain the most extensively validated in AA, while more recently proposed indices—such as the monocyte-to-lymphocyte ratio (MLR), systemic immune-inflammation index (SII), systemic inflammation response index (SIRI), and pan-immune-inflammation value (PIV)—show increasing promise for preoperative risk stratification. Several studies have reported their potential usefulness not only in diagnosing appendicitis but also in distinguishing CAA from UAA [10,13,26,27,28,29].

In our cohort, PLR values were significantly higher in cases classified as complicated based on intraoperative findings, but this difference was attenuated and no longer statistically significant when complicated appendicitis was defined solely by histopathological NUP status. These results partly contrast with those of Zhang et al., who divided adult appendicitis cases into uncomplicated and complicated according to pathology and found that PLR was significantly higher in complicated appendicitis and contributed to the identification of these patients together with other laboratory parameters [30]. Similarly, Yıldırım et al. reported that PLR was significantly higher in adults with complicated appendicitis and independently predicted complications, using a classification based on intraoperative findings combined with pathology reports [31]. One possible explanation is that PLR may reflect the overall systemic inflammatory burden and macroscopic disease severity encountered intraoperatively more than the presence of microscopic wall necrosis alone; thus, its discriminatory power is greater when complicated appendicitis is defined by clinical–surgical criteria (perforation, abscess, plastron) than when a more restricted histological definition (NUP) is applied.

In our cohort, SII and SIRI were significantly higher in complicated than in uncomplicated appendicitis, with AUC values around 0.688–0.707. This pattern is in keeping with adult studies by Ghoncheh et al., Yildiz et al., Montero et al., Berhuni et al., which also found higher SII/SIRI in complicated disease [10,11,32,33]. Our AUCs were slightly higher than those reported by Berhuni et al. for SII, but the direction and magnitude of the association were similar [11].

Among the evaluated indices, the Pan-Immune Inflammation Value (PIV) remains the least investigated marker in the context of AA. Only a few recent studies, such as those by Saridas et al. (2024) and Yarkaç et al. (2025), have assessed its diagnostic utility in differentiating complicated from uncomplicated cases, reporting acceptable but modest discriminatory performance [13,34]. In comparison, the monocyte-to-lymphocyte ratio (MLR) has been explored more frequently; however, a considerable proportion of this literature focuses on pediatric or mixed-age cohorts, while evidence in adult populations remains scarce. Although both indices show potential as accessible and cost-effective markers of systemic inflammation, their exact diagnostic and prognostic roles in appendicitis are not yet clearly established [8,13,14,34,35,36,37,38]. Further large-scale, prospective studies are warranted to validate their clinical relevance and to determine standardized cut-off values applicable across age groups and healthcare settings.

In our study, all six indices demonstrated acceptable diagnostic power in differentiating CAA from UAA, regardless of whether intraoperative findings or histopathological confirmation was used as the reference standard. This observation is noteworthy because intraoperative and histopathological assessments are not always concordant, as previously highlighted by Bolmers et al. and others, who described meaningful rates of misclassification and over- or underestimation of disease severity [39]. Beyond diagnostic value, our results—and those of recent reports—suggest that higher PIV levels correlate with extended length of hospital stay (LOS) and higher inpatient costs, indicating that this parameter may also serve as a prognostic marker reflecting systemic inflammatory burden [12,40,41]. As these indices can be automatically calculated from a routine complete blood count, their integration into emergency diagnostic workflows is both practical and cost-neutral, making them attractive tools for rapid preoperative triage in resource-limited settings.

While hematologic immune–inflammatory indices offer valuable adjunctive information, they should not be used in isolation to guide the management of AA. Given that all AUCs for these indices fell within the fair range (0.66–0.72), they should be regarded as supportive tools that complement, rather than replace, imaging findings and established clinical scores in the assessment of suspected complicated appendicitis. Their value may be greatest in resource-limited settings, where access to advanced imaging or formal scoring systems is restricted and inexpensive laboratory markers can provide additional risk stratification.

The decision between operative and non-operative treatment must remain a multifactorial process, integrating, where possible, clinical presentation, imaging findings, patient comorbidities, and shared decision-making with the patient [2,12,19]. Laboratory-derived indices such as NLR, PLR, SII, SIRI, and PIV may enhance diagnostic and prognostic precision, but they cannot substitute for thorough clinical evaluation [10,12,34,42]. Instead, these biomarkers should be viewed as complementary components within a broader diagnostic framework that incorporates radiological and clinical risk assessment tools. Emerging evidence suggests that combining laboratory indices with imaging-based or metabolic markers could yield future decision-support models capable of safely identifying candidates for non-operative management [12,42,43,44,45,46]. Such integrated approaches represent the next step toward individualized, evidence-based triage in AA care.

This study has certain limitations. It was conducted retrospectively at a single center, which may limit the generalizability of the results. All included patients were surgically treated, reflecting our institutional practice in which non-operative management of AA is not routinely applied. While this approach ensures histopathological confirmation for all cases, it also introduces selection (spectrum) bias, as our findings pertain only to patients already selected for appendectomy and may not fully apply to centers that routinely offer antibiotic-only treatment. Consequently, we could not evaluate the ability of these indices to identify candidates for non-operative management or to predict failure of conservative therapy. Future prospective, multicenter studies in settings where both NOM and surgery are used are needed to determine how these markers can be integrated into decision-making algorithms that triage patients between operative and non-operative pathways. Additionally, inflammatory marker levels can be influenced by comorbidities, medications, and the timing of sample collection. Future prospective, multicenter studies are needed to validate these findings and to define standardized cut-off values applicable across diverse emergency settings. Immunosuppressed patients (e.g., chronic corticosteroid or other immunosuppressive therapy) and patients who had received antibiotics before admission were not excluded systematically and were analyzed together with the overall cohort, reflecting routine emergency practice, but we recognize the potential effect of these factors on the inflammatory indices. Additionally, we did not include CT-measured appendiceal diameter in the models for complicated appendicitis because CT was not systematically performed in all patients; therefore, some residual confounding by imaging findings cannot be fully excluded.

## 5. Conclusions

Our findings demonstrate that emerging immune–inflammatory indices such as NLR, MLR, SII, SIRI and PIV show moderate ability to distinguish complicated from uncomplicated acute appendicitis, and that higher PIVs are independently associated with longer hospital stay and higher inpatient costs. Because these markers can be derived from routine blood tests, they may serve as easily accessible adjuncts to preoperative risk stratification, particularly in settings with limited imaging resources. However, they should be used in conjunction with clinical assessment and imaging findings, rather than as standalone diagnostic tests. Prospective, multicenter studies are needed to validate these findings and to determine whether incorporating these indices into clinical decision-making can improve resource utilization and patient outcomes.

## Figures and Tables

**Figure 1 diagnostics-16-00021-f001:**
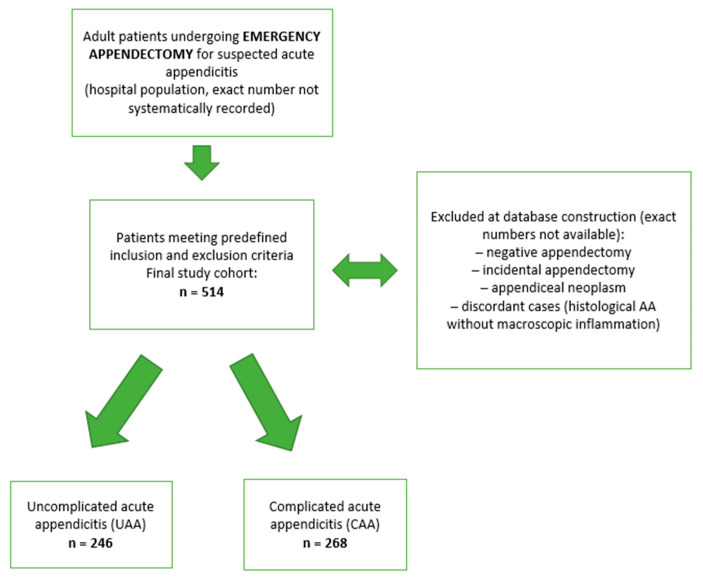
Flow diagram of patient selection and stratification.

**Figure 2 diagnostics-16-00021-f002:**
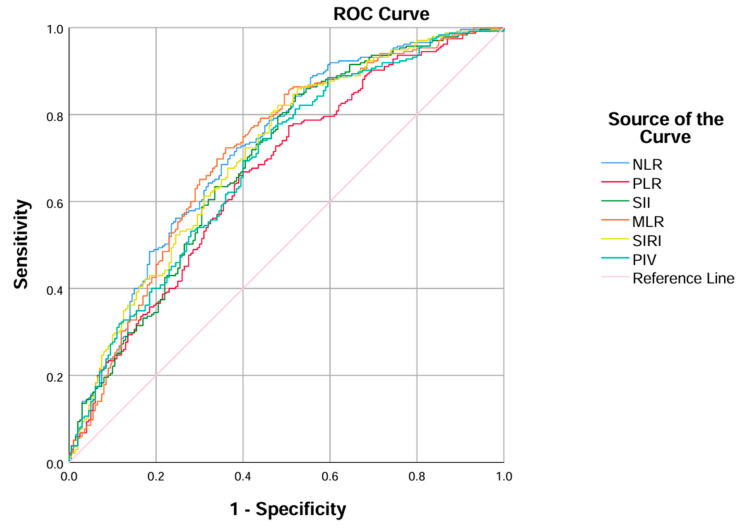
ROC curves of immune–inflammatory indices (NLR, PLR, SII, MLR, SIRI, PIV) for identifying complicated appendicitis.

**Table 1 diagnostics-16-00021-t001:** Perioperative Characteristics and Inflammatory Indices by Appendicitis Severity (Complicated vs. Uncomplicated).

Variables		*n*	Mean	Standard Deviation (±)	*p* Value
AGE (years)	CAA	268	46.28	18.061	0.0001 *
	UAA	246	35.03	14.661	
LEU (×10^9^/L)	CAA	247	14.852	4.567	0.0001 *
	UAA	233	13.073	4.263	
NEU (×10^9^/L)	CAA	247	12.262	4.257	0.0001 *
	UAA	232	10.138	4.303	
PLT (×10^9^/L)	CAA	246	248.024	73.706	0.249
	UAA	232	255.068	59.239	
MONO (×10^9^/L)	CAA	235	0.978	0.413	0.001 *
	UAA	201	0.858	0.337	
CRP (mg/L)	CAA	20	189.529	83.808	0.0001 *
	UAA	18	38.894	51.484	
Operative time (min)	CAA	267	79.78	26.25	0.0001 *
	UAA	246	60.75	20.999	
**Variables**		* **n** *	**Median**	**IQR**	***p*** **Value**
NLR	UAA	233	5.297	2.991–8.686	<0.0001 *
	CAA	246	9.104	6.05–13.136	
PLR	UAA	232	136.575	97.375–197.306	<0.0001 *
	CAA	246	180.352	139.377–249.446	
SII	UAA	232	1350.638	671.011–2341.4	<0.0001 *
	CAA	246	2158.381	1456.713–3107.087	
MLR	UAA	232	0.43	0.306–0.662	<0.0001 *
	CAA	236	0.703	0.5–0.974	
SIRI	UAA	201	4.537	2.049–7.944	<0.0001 *
	CAA	235	8.099	5.286–13.421	
PIV	UAA	200	1205.89	497.149–2043.101	<0.0001 *
	CAA	235	1943.603	1299.709–3277.582	
LYM (×10^9^/L)	UAA	233	1.86	1.34–2.475	0.0001 *
	CAA	246	1.31	0.93–1.707	
CT (diameter, mm)	UAA	77	10	8.15–12.00	<0.0001 *
	CAA	114	13	10.00–15.00	
LOS (days)	UAA	246	3	2.00–4.00	<0.0001 *
	CAA	268	4	3.00–6.00	
**Variables**		**CAA**		**UAA**	***p*** **Value**
Sex	Male *n*(%)	164 (56.6)		126 (43.4)	0.023 *
	Female *n* (%)	104 (46.4)		120 (53.6)	

NLR—neutrophil-to-lymphocyte ratio; PLR—platelet-to-lymphocyte ratio; SII—systemic immune-inflammation index; MLR—monocyte-to-lymphocyte ratio; SIRI—systemic inflammation response index; PIV—pan-immune-inflammation value; UAA—uncomplicated acute appendicitis; CAA—complicated acute appendicitis; IQR—interquartile range; LEU—leucocyte; NEU—neutrophil; PLT—platelet; MONO—monocyte; LYM—lymphocyte; CRP—c-reactive protein; *—significant *p*-value, LOS—length of stay.

**Table 2 diagnostics-16-00021-t002:** Association of Immune–Inflammatory Indices with histological all destruction (NUP Status).

Indices	NUP+	Median	IQR		*p* Value	Z
NLR	0	6.159	3.949	11.179	0	−3.762
	1	8.58	5.652	11.818		
PLR	0	156.949	120.704	215.763	0.235	−1.186
	1	171.612	126.486	230.985		
SII	0	1600.962	1015.907	2785.409	0.01	−2.584
	1	2030.899	1394.221	2952.743		
MLR	0	0.53	0.354	0.791	0.001	−3.222
	1	0.644	0.433	0.913		
SIRI	0	5.671	2.563	9.929	0.002	−3.075
	1	7.348	4.474	11.654		
PIV	0	1484.332	626.736	2457.868	0.019	−2.34
	1	1767.877	1110.886	2899.216		

NUP+—necrosis, ulceration, perforation on the histopathological letter, IQR—interquartile range, Z—standardized test statistic (Mann–Whitney U test).

**Table 3 diagnostics-16-00021-t003:** Diagnostic performance of immune-inflammatory indices for identifying complicated appendicitis.

Indices	AUC	Youden Index	Sens	Spec	CUTOFF	PPV	NPV	Accuracy
NLR	0.719	0.347	0.866	0.481	5.05	0.637	0.772	0.678
PLR	0.664	0.281	0.768	0.513	138.52	0.625	0.676	0.644
SII	0.688	0.329	0.837	0.509	1300.39	0.635	0.740	0.669
MLR	0.712	0.363	0.72	0.644	0.5375	0.702	0.663	0.684
SIRI	0.707	0.339	0.821	0.517	4.5957	0.665	0.712	0.681
PIV	0.687	0.301	0.766	0.535	1279.43	0.659	0.660	0.659

AUC—area under the curve, Sens—sensitivity, Spec—specificity, PPV—positive predictive value, NPV—negative predicting value.

**Table 4 diagnostics-16-00021-t004:** Independent Predictors of Complicated Appendicitis (CAA) and Histopathological Severity (NUP+).

CAA Group								
Step	Variable	B	S.E.	Wald	*p* (Sig.)	Exp(B)	95% CI Lower	95% CI Upper
1	NLR	0.125	0.022	33.853	<0.001	1.134	1.087	1.183
2	NLR	0.078	0.028	7.65	0.006	1.081	1.023	1.142
	MLR	1.029	0.427	5.808	0.016	2.798	1.212	6.459
**NUP+** **Group**								
**Variable**	**B**	**S.E.**	**Wald**	**df**	**Sig.**	**Exp(B)**	**95% CI Lower**	**95% CI Upper**
MLR	0.542	0.259	4.377	1	**0.036**	**1.719**	**1.035**	**2.856**
Constant	−0.318	0.201	2.5	1	0.114	0.727	—	—

CAA—intraoperative diagnosis of complicated appendicitis; NUP+—necrosis, ulceration, perforation present on the histopathological data, NLR—neutrophil-to-lymphocyte ratio, MLR—monocyte-to-lymphocyte ratio, B—regression coefficient, S.E.—standard error of the coefficient, Wald—Wald χ^2^ test statistic, *p* (Sig.)/Sig.—*p*-value (statistical significance), Exp(B)—exponentiated coefficient, i.e., odds ratio, 95% CI—95% confidence interval, df—degrees of freedom.

**Table 5 diagnostics-16-00021-t005:** Associations of Inflammatory Indices with Length of Stay and Hospital Costs: Hierarchical Regression Results.

Outcome	Model Summary	Predictor	B	β	t	*p* Value	VIF
LN_LOS	R^2^ = 0.382 (Adj R^2^ = 0.347) ΔR^2^ = 0.137, *p* < 0.001	PLR	0.0004	0.094	1.04	0.303	1.19
		PIV	0.0008	0.326	3.36	0.001	1.21
LN_COSTS	R^2^ = 0.329 (Adj R^2^ = 0.292) ΔR^2^ = 0.160, *p* < 0.001	PLR	0	0.155	1.63	0.107	1.19
		PIV	0	0.315	3.31	0.001	1.21

Dependent variables are ln-transformed. All models adjusted for age, sex, and residential status (Block 1). ΔR^2^ represents additional variance explained after inclusion of PLR and PIV (Block 2), LN_LOS—natural logarithm of length of hospital stay, , LN_COSTS—natural logarithm of total hospital costs, PLR—platelet-to-lymphocyte ratio, PIV—pan-immune-inflammation value, R^2^—coefficient of determination, Adj R^2^—adjusted coefficient of determination, ΔR^2^—change in R^2^ (additional explained variance), B—unstandardized regression coefficient, β—standardized regression coefficient, t—t-statistic, VIF—variance inflation factor.

## Data Availability

The raw data supporting the conclusions of this article will be made available by the authors on request.

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
