# Peer review of "The Role of Emerging Immune-Inflammatory Indexes in the Preoperative Differentiation of Complicated and Uncomplicated Acute Appendicitis: A Single-Center Retrospective Analysis"

_diagnostics, 2025, doi:10.3390/diagnostics16010021_

Round 1

Reviewer 1 Report

Comments and Suggestions for Authors

In their retrospective, single-center study, the authors assessed the accuracy of several inflammatory biomarkers to differentiate between complicated and uncomplicated acute appendicitis, using commonly available inflammatory indices. The topic holds clinical significance, the cohort size is large (n=514), and the overall organization and scientific writing are clear. The study offers valuable evidence supporting the diagnostic and prognostic usefulness of immune–inflammatory markers such as NLR, MLR, SIRI, SII, and PIV. The authors concluded that easily calculated inflammatory markers can function as useful adjuncts for preoperative stratification of acute appendicitis, aiding timely decision-making and helping to make emergency surgical care more cost-effective.

  1. The introduction should expand on other markers or diagnostic scores that have already been proven to differentiate simple from complicated appendicitis, such as hyponatremia (doi: 10.3390/children9071070), procalcitonin, delta neutrophil index (DNI) (doi: 10.1515/med-2025-1308), or hyperbilirubinemia (doi: 10.7759/cureus.93197), and established clinical scores like the Appendicitis Inflammatory Response (AIR) score (doi: 10.3390/children8040309), which have shown predictive value for complicated disease. Including these well-validated parameters and references would better highlight the novelty of these markers within the landscape of existing diagnostic tools and emphasize its potential added diagnostic benefit.
  2. Additionally, to stay current, recent studies have shown that salivary biomarkers have become a promising, non-invasive addition to existing diagnostic methods, offering practical benefits like easy collection, no pain, and less stress, especially for children, although further validation is still necessary (doi: 10.3390/children12101342). This should also be included in the introduction.
  3. The authors presented inclusion criteria, exclusion criteria should be presented, as well.
  4. The authors did not explicitly identify the primary and secondary outcomes of the study. Please include a separate paragraph in the methodology section - outcomes of the study.
  5. The study design should be described in more detail. Specifically, the authors should specify the exact variables measured in each patient. They also need to list which laboratory, demographic, or clinical data were recorded, etc.
  6. The issue of selection bias in this study should be clarified. The results show that all patients in the cohort underwent surgery, with cases managed non-operatively excluded. This could limit the clinical relevance in settings where NOM is routine. Please explicitly recognize and elaborate on how this impacts external validity and the role of markers in decision-making algorithms.
  7. The authors did not specify whether blood tests were consistently collected at admission, before administering analgesia or antibiotics, or right before surgery. These factors can affect inflammatory marker levels. Clarification is necessary to ensure the values' interpretability.
  8. Although NLR, MLR, and SIRI performed better than PLR, all AUCs remained within the range of fair diagnostic performance (0.66–0.72). The discussion should highlight that these markers support but do not outperform imaging or clinical scoring tools.
  9. The authors presented optimal cut-offs determined by Youden index, but it is unclear whether these were validated (through bootstrapping or cross-validation). Clarify whether these cut-offs are exploratory or recommended for clinical use.
  10. In Table 1, ensure consistent decimal formatting and alignment for easier clinical interpretation. Additionally, delete the column with Z-scores and format the table according to journal guidelines (see previously published articles). The current table is cluttered and difficult to interpret. The same applies to other tables in the manuscript.
  11. The conclusions would be stronger if they explicitly state that these indices should be used alongside imaging and clinical examination, not as standalone tests.
  12. The manuscript would benefit from the inclusion of a flowchart outlining patient selection and stratification (especially exclusions), as this would improve overall clarity and enhance readability.
  13. The manuscript needs English language editing because of several grammar, syntax, and phrasing issues that impact readability. A professional language review is recommended.

Reviewer 2 Report

Comments and Suggestions for Authors

Dear authors,

Thank you for submitting this paper. While few similar studies have been conducted so far the topic remains clinically relevant. However, some issues remain which I will state below:

1) In study population, one of the inclusion criteria is intraoperative and histopathological confirmation of acute appendicitis. What was the reasoning for excluding the patients which had histopathological confirmation but not intraoperative? Were there any other exclusion criteria such as immunosuppressed patients, patients receiving antibiotics in period before being admitted to the hospital and such.

2) While I support usage only of patient that have recieved surgical treatment, comparing the values of these markers against non appendicitis patients would add value

3) ALL tables require reformatting/styling, as I don't find the current format acceptable. Some abbreviations should be named differently such as "OP_TIME". Where applicable you should add unit of measurement in tables like for AGE (years), CRP... I would suggest adding lenght of hospital stay in table 1.

4) I find a bit troubling that you used intraoperative diagnosis for definition of complicated appendicitis in calculating ROC curves especially while at the same time you're discussing the potential discrepancies between intraoperative finding and histopathological findings. I would suggest performing ROC curve analysis for each. Please expand on your reasoning for that

5) In table 3 you report all Youden index values as >1 while that metric ranges from -1 to 1.

6) Tone down claims such as "... indices demonstrated robust diagnostic power" considering the values reported

7) in this dataset age, sex and appendix CT diameter differ between groups which is why I would recommend constructing multivariable models with these features as this could lead to false attribution of predictive capacity of these indices

8) some variables have missing values which is not described accurately

9) date of approval should be revised - "118.11.2024."

10) discussion should be revised to focus on study results more and comparison with already conducted research

Reviewer 3 Report

Comments and Suggestions for Authors

Authors aimed to investigate the diagnostic accuracy and prognostic relevance of novel immune-inflammatory indices (NLR, PLR, MLR, SIRI, SII, and PIV) in acute appendicitis in this study.

Acute appendicitis is the most common surgical emergency. Laboratory values that can help diagnose the disease and determine whether the appendix has become complicated are of great importance. The ability to diagnose the disease using only clinical findings and laboratory values, without the need for advanced radiological imaging methods such as ultrasonography and computed tomography, is a significant advantage. This is particularly important for physicians in rural areas and primary care settings who may have limited access to imaging methods. Laboratory tests such as hemogram and biochemistry are inexpensive, are easily accessible, and provide quick results. Therefore, this study, which emphasizes the importance of laboratory values in diagnosing acute appendicitis and determining its complicated status, is valuable and will make a significant contribution to the literature. Icongratulate the authors on their successful work. Some situations in the study need to be reviewed and I hope that our comments will contribute to author’s study.

Comments to Authors.

1: The word 'novel' in the study's title and within the body of the work, specifically referring to '….novel inflammatory indexes', should be reconsidered. Since these markers have been used in many previous studies, the term 'novel' might be too ambitious. I believe the word 'novel' should be removed. However, if the authors, after re-evaluation, still deem the use of the word correct, their decision will ultimately be the most appropriate one

2: Keywords: The list of keywords should be revised to reduce redundancy. Consider replacing the enumerated specific indices (preoperative stratification, pan-immune-inflammation value, systemic inflammation response index, systemic immune inflammation index, monocyte-to-lymphocyte ratio, neutrophil-to-lymphocyte ratio, platelet-to-lymphocyte ratio) with the broader term "Laboratory Parameters". Furthermore, I suggest deleting "risk factors" as it does not reflect the primary objective of this study. (Please ensure appropriate MeSH terms are utilized.)

3: The sentence in the Results section: 'CT-measured appendiceal diameter was greater in complicated cases (median 13 mm 161 [IQR 10–15] vs. 10 mm [8.15–12]; p < 0.0001).' is unnecessary and not relevant to the study's objective. This sentence should be removed. Similarly, the operation time and CT findings should be removed from the tables.''

4: The typos in Table 1 should be corrected. The abbreviation for Platelet should be Plt, and Monocyte should be abbreviated as Mono

Discussion

5-Abbreviations must be defined by writing out the full term followed by the abbreviation in parentheses when first mentioned. In subsequent uses, either the abbreviation or the full term should be used. Therefore, the writing of 'Acute appendicitis (AA)' in the Discussion section is incorrect (assuming it was defined earlier). This and similar instances of incorrect abbreviation usage throughout the manuscript must be corrected.

6-In discussion ‘’ The presence of an appendicolith further complicates therapeutic decision-making in acute appendicitis. Several randomized and observational studies have shown that non operative management fails significantly more often when an appendicolith is present, leading to higher early recurrence and complication rates [5,21,22]. In the CODA trial, 41% of antibiotic-treated patients with an appendicolith required appendectomy within 90 days, compared with 25% of those without [5,23]. Subsequent cohort analyses and meta303 reviews confirmed that appendicoliths nearly double the risk of treatment failure and re current inflammation following antibiotic therapy [23–25]. While the optimal management of uncomplicated cases with an appendicolith remains debated, urgent surgical intervention for CAA is universally recognized as essential to prevent perforation, postoperative morbidity, and excess healthcare costs [2,20]. ‘’ I believe this section is unnecessarily included. It is not consistent with the study's aim. I recommend removing this part from the study

7-….. only in diagnosing appendicitis but also in distinguishing CAA from UAA [10,13,26–28]. I recommend that you also add these current and important studies in this field to the references in the Discussion section (DOI: 10.14744/tjtes.2020.69195,, doi: 10.1111/ans.17443)

Conclusion Section

8-‘’Moreover, the observed associations with hospital costs and length of stay underscore their broader relevance for optimizing resource allocation and improving overall outcomes. Further prospective validation is warranted to consolidate their role as cost-effective, rapid tools in emergency surgical decision-making.’’ This finding is not a result derived from your study. Although the use of these sentences is not technically incorrect, I do not think it is appropriate for your study to emphasize them in the Results section. The Results section should be rewritten. These studies may be helpful and serve as a guide for you:(DOI: 10.14744/tjtes.2020.69195,, doi: 10.1111/ans.17443)

Round 2

Reviewer 1 Report

Comments and Suggestions for Authors

The revised manuscript has been substantially improved. All previously raised concerns have been addressed thoroughly and appropriately. The language quality is also improved. I appreciate the authors’ careful and comprehensive revisions.